# Validation of a Culturally Relevant Snakebite Envenomation Clinical Practice Guideline in Brazil

**DOI:** 10.3390/toxins14060376

**Published:** 2022-05-28

**Authors:** Gisele dos Santos Rocha, Altair Seabra Farias, João Arthur Alcântara, Vinícius Azevedo Machado, Felipe Murta, Fernando Val, Joseir Saturnino Cristino, Alícia Cacau Santos, Mena Bianca Ferreira, Leonardo Marques, Yasmim Vieira Rocha, André Sachett, Mailma Costa Almeida, Aline Alencar, Lisele Brasileiro, Érica da Silva Carvalho, Pedro Ferreira Bisneto, Marcus Lacerda, Anna Tupetz, Catherine A. Staton, João R.N. Vissoci, Elizabeth Teixeira, Charles J. Gerardo, Fan Hui Wen, Jacqueline Sachett, Wuelton Monteiro

**Affiliations:** 1Escola Superior de Ciências da Saúde, Universidade do Estado do Amazonas, Manaus 69065-001, Brazil; grocha@uea.edu.br (G.d.S.R.); asfarias@uea.edu.br (A.S.F.); arthuralcant@hotmail.com (J.A.A.); vmachado@uea.edu.br (V.A.M.); felipemurta87@yahoo.com.br (F.M.); ffaval@gmail.com (F.V.); joseysaturnino@gmail.com (J.S.C.); aliciacacau22@gmail.com (A.C.S.); mnbiancapaiva@gmail.com (M.B.F.); leolincoln.lincoln@hotmail.com (L.M.); yvdr.enf@uea.edu.br (Y.V.R.); andre.sachett@gmail.com (A.S.); mcalmeida@uea.edu.br (M.C.A.); alencaraline.med@gmail.com (A.A.); liselebrasileiro@hotmail.com (L.B.); carvalhouea@gmail.com (É.d.S.C.); pedro.fbisneto@hotmail.com (P.F.B.); marcuslacerdabr@gmail.com (M.L.); etfelipe@hotmail.com (E.T.); jac.sachett@gmail.com (J.S.); 2Diretoria de Ensino e Pesquisa, Fundação de Medicina Tropical Dr. Heitor Vieira Dourado, Manaus 69040-000, Brazil; 3Programa de Pós-Graduação em Zoologia, Universidade Federal do Amazonas, Manaus 69067-005, Brazil; 4Instituto Leônidas & Maria Deane, Fundação Oswaldo Cruz, Manaus 69057-070, Brazil; 5Division of Emergency Medicine, Duke University School of Medicine, Durham, NC 27708, USA; anna.tupetz@duke.edu (A.T.); catherine.staton@duke.edu (C.A.S.); jnv4@duke.edu (J.R.N.V.); charles.gerardo@duke.edu (C.J.G.); 6Instituto Butantan, São Paulo 05503-900, Brazil; fan.hui@butantan.gov.br; 7Diretoria de Ensino e Pesquisa, Fundação Alfredo da Matta, Manaus 69065-130, Brazil

**Keywords:** snakebite, antivenom, health education, clinical practice guideline, validation

## Abstract

Snakebite envenoming (SBE) is a neglected tropical disease with significant global morbidity and mortality. Even when antivenom is available in low-resource areas, health workers do not receive adequate training to manage SBEs. This study aims to develop and validate a clinical practice guideline (CPG) for SBE management across Brazil. A panel of expert judges with academic and/or technical expertise in SBE management performed content validation. The content validity index (CVI) score was 90% for CPG objectives, 89% for structure and presentation and 92% for relevance and classified the CPG as valid. A semantic validation was performed by analyzing focus group discussions with doctors and nurses from three municipalities of the Brazilian Amazon, after a 5-day meeting during which the CPG was presented. Two central themes emerged: knowledge acquired during the meeting and recommendations for improving the CPG. Based on these results, the CPG was revised into a final version. This study presents the successful development and validation process of a CPG for SBE management, which is targeted to a specific low-resource, high-burden setting. This development and validation process can be adapted to other settings and/or other neglected tropical diseases.

## 1. Introduction

Snakebite envenoming (SBE) is considered a neglected tropical disease and public health problem that is more prevalent in underdeveloped, tropical countries, and about 2.5 million envenomings and at least 100,000 deaths occur each year [1,2]. Of the victims that survive, around 400,000 suffer from long-term physical and psychological sequelae, which impact their emotional, social and financial well-being. Due to underreporting in underdeveloped countries, these figures underestimate the true impact of SBEs [1,2]. In 2019, 30,482 cases of snakebites were reported in Brazil. Over 44% of snakebites occurred in the Amazon region of Brazil, a proportion which is five times higher than that of any other state, despite this region having only 8.7% of the country’s total population [3].

In 2018, the World Health Organization (WHO) launched a strategy to reduce mortality and disability caused by SBE by half by 2030. This strategy will include ensuring safe, accessible and effective treatment; empowering communities to be proactive in preventing bites; improving access to treatment; and strengthening local health systems to achieve better outcomes for patients [4].

For the management of SBEs, antivenom is the only treatment with proven efficacy [1,2]. Thus, quick access to this treatment is crucial to avoid acute complications, sequelae and deaths. In practice, however, equitable access to antivenom is limited by its uneven distribution and availability [5,6]. In the Brazilian Amazon region, each municipality has only one hospital that provides SBE antivenom treatment. As there is no antivenom available in the rural health units near where most SBEs occur, hundreds of thousands of people living in rural areas, riverine communities and indigenous areas are denied the possibility of timely and adequate treatment. Therefore, the expansion of antivenom therapy to health facilities in these remote areas would give these high-risk populations access to appropriate treatment. Yet, in snakebite-endemic countries, evidence suggests that health workers lack sufficient knowledge of SBE management and feel insecure when treating SBEs, which is driven by limited SBE training of these professionals during their clinical education [7,8]. In the presence of a validated protocol, health professionals other than physicians are able to prescribe timely life-saving medications/immunobiologicals, which already happens in cases of malaria, leishmaniasis, tuberculosis, etc. [9].

In order to ensure adequate and quality care, it is therefore critical, while improving access to antivenom, to simultaneously create and validate antivenom administration protocols. The training of medical personnel and the development of standard treatment protocols are thus priorities for adequate SBE management. As such, the present study aims to develop a clinical practice guideline (CPG) that guides health professionals in the use of antivenom and SBE management in Brazil. In addition to current antivenom care guidelines, we assembled practical ancillary treatment solutions including first aid, wound care and pain management.

## 2. Materials and Methods

The study follows a mixed-methods approach to develop and validate a CPG for SBE treatment in the Brazilian context. An overview of the methods is presented in Figure 1.

### 2.1. Research Team and Reflexivity

A steering group composed of three PhD-level researchers with a track record (F.H.W., J.S., W.M.) in the field of clinical research and care of SBE patients (two from *Fundação de Medicina Tropical Dr. Heitor Vieira Dourado* (FMT-HVD), a tertiary healthcare unit that treats SBE patients in the western Amazon region, and one from the *Instituto Butantan* (IBU), the major antivenom producer in the country, which also treats SBE patients in the city of São Paulo) was established to elaborate the first version of the CPG and to coordinate the validation process.

Two female researchers and one male researcher moderated the focus groups as part of the semantic validation by end-users. Two moderators have Master of Science degrees and one has a PhD degree, though all three are trained in qualitative research methods. The study team included five physicians with extensive experience in SBE research, two licensed physical therapists (data specialists), seven nurses (one with extensive experience in SBE research, two care package validation specialists and four specialists in qualitative research), three clinical laboratory scientists, three educators (specialists in qualitative research), two psychologists (data specialists), one herpetologist, one clinical epidemiologist and two clinical research experts.

### 2.2. Care Practice Guideline Development

The plan for the development of the CPG began with the identification of existing related guidance for SBE management in Brazil and the definition of the scope of the CPG, which included objectives and potential sections.

The Brazilian Ministry of Health has an official guideline with general recommendations for the management of SBEs and envenomations by other venomous animals [10]. This guideline presents the epidemiological and general clinical aspects of envenomations by the four genera of snakes of medical importance in the country (*Bothrops*, *Lachesis*, *Crotalus* and *Micrurus*), as well as the available antivenoms and their dosages according to SBE severity. The Brazilian Ministry of Health (MoH) guideline was originally created based on expert opinion considering the experience of patients treated in hospitals, under medical supervision. The guideline does not address pre-hospital care and first aid; antivenom storage, preparation and administration; wound care; ancillary treatment of different local and systemic manifestations; referral to higher complexity healthcare services; discharge criteria; clotting time procedure; and reporting of cases to epidemiological surveillance systems. A literature search via MEDLINE and LILACS on 31 January 2020 (in Portuguese and English), which used the descriptors (snakebite OR snake bite OR snake envenomation OR ophidian accident OR ophidian bite OR ophidian envenomation OR snake accident) AND (care package OR treatment guideline OR treatment package) AND Brazil, did not provide any additional published guidelines for the treatment of SBEs in Brazil.

The first version of the CPG was written with 13 chapters (Figure 1). It is noteworthy that in the CPG validated in this study, first aid and pre-hospital care procedures that can be performed by paramedic staff are included. Detailed information on which antivenom treatment can be performed in community health centers depending on the local policies for snakebite treatment and future plans for decentralization of antivenom treatment in the Brazilian Amazon is also included [3]. Antivenom dosages are presented using the same standards as the MoH recommendations since clinical studies performed in the Amazon have demonstrated that the dosages of the standard antivenoms are effective in reversing the effects of envenoming on hemostasis [11].

### 2.3. Content Validation by Expert Judges

Professors of graduate programs in tropical medicine and clinical toxicology were selected and invited to participate as expert judges for this study, as long as they act in lines of research related to SBEs. Health staff trained in the care of SBE patients in tertiary hospitals were also invited to participate. Initially, researchers and professionals identified from scientific publications retrieved from MEDLINE with publications within the last 5 years were invited. Additionally, the indication of experienced health professionals (physicians, registered nurses and other health professionals) by these researchers was requested. Some hospitals were also contacted to obtain recommendations. The aim was to obtain a culturally and geographically diverse sample that could evaluate the validity and reliability of the revised care information package in Brazil. The committee was further divided into two groups: Group 1 (G1) (expert judges with academic experience) and Group 2 (G2) (health professionals with technical expertise in SBE management). The inclusion of the judges in G1 was established by the achievement of a score of five points in the following items: professional training, experience and scientific publications in the field of SBEs). For G2, the same criteria were considered, with a requirement of reaching a minimum score of three points, as well as having at least 2 years of experience in the treatment of SBE patients [12] (Table 1).

The invitation was made via an email sent to the potential judges, which also had the first version of the CPG attached in portable document format (PDF) and a link (Google forms) with the informed consent form. In case of no response within 20 days after the invitation, the judge was not included. Judges were excluded if they did not return the fully completed and legible instrument within 20 days after the validation instrument was sent.

The validation instrument used was a questionnaire organized into three parts: the judge’s details; orientations for filling in the questionnaire; and three blocks with items on the (1) objectives, (2) structure and presentation and (3) relevance of the CPG. These items were evaluated using a Likert-type scale, with space for comments in each block. Each judge was able to express their assessment in the three blocks using scores of 1 to 4 (1, fully adequate; 2, adequate; 3, partially adequate; and 4, inadequate). In addition to this validation instrument, the judges were also asked to send any comments they considered pertinent to the improvement of the care information package. A second version of the care information package was produced based on the scores obtained after applying the validation instrument and accepting the judges’ comments and opinions.

### 2.4. Semantic Evaluation by Target Users

In order to validate the care information package with the target audience, doctors and nurses working in the health service of the municipalities of Careiro da Várzea, Ipixuna and Boa Vista do Ramos, in the state of Amazonas, were invited to a 5-day (28 June to 2 July 2021) meeting in Manaus. The professionals were appointed by the municipal health managers and agreed to participate. The second version of the care information package was presented to them through lectures and discussions of real clinical cases. The professionals who completed at least 75% of the meetings were invited to participate in focus groups with a maximum of 10 participants per group. Participants who agreed to participate signed a consent form after an explanation of the study’s objectives and procedures.

Prior to the sessions, areas of interest to be discussed during the focus groups were specified by the research group, which was led by the steering group. A semi-structured interview guide was developed and piloted on a smaller sample of volunteers (Appendix A). Each focus group was conducted by an experienced moderator in a comfortable and quiet room where the moderator, the observers and the participants were accommodated. The research team had no prior relationships with any of the participants. The team introduced themselves personally to the participants at the beginning of the focus groups and provided a short background on their role in the study. The focus groups lasted an average of 60 min and were recorded using video and audio devices, and field notes were taken by the interviewer and by the observers. The interviews were recorded and transcribed, and the de-identified versions were subsequently loaded on the MAXQDA 20 program.

After the evaluation of the CPG’s semantics by the target users, its final version was elaborated by analyzing the suggestions in order to improve content and structure via a more comprehensive language and better format style. The final version of the CPG was revised by a Portuguese language teacher.

### 2.5. Data Analysis

Descriptive statistics were used for characterizing study participants and to obtain the content validity index (CVI) for the expert evaluation of the CPG content [13]. The CVI was calculated by the sum of agreement of the items marked as “totally adequate” and “adequate”, divided by the total of responses, multiplied by 100. For the protocol to be considered content-validated, a CVI equal to or greater than 0.80 was established for all the analysis blocks. Items that received “partially adequate” or “inadequate” evaluations were reviewed and suggested changes were added. Results from judges with academic expertise and healthcare professionals with field expertise were combined for analysis.

The qualitative analysis was carried out through a thematic content analysis, with the elaboration of categories that emerged during the analysis process after the previous reading of the transcripts. Data source triangulation (transcriptions, observer notes) was used during the analysis, and the codebook was created in MAXQDA software by two researchers (W.M.M. and F.M.) independently. The discrepancies in the coding were resolved with discussion to reach a consensus.

Results were presented according to the Consolidated Criteria for Reporting Qualitative Research (COREQ) (Appendix A).

## 3. Results

### 3.1. Characterization of Expert Judges

Invitation letters were sent to 51 potential judges, which yielded 24 acceptances. In all, 20 returned the form duly filled out and were analyzed; 12 judges had academic expertise in clinical research of SBEs and 8 had SBE management expertise. The age of the judges ranged from 29 to 66 years, and professional experience ranged from 3 to 40 years. Judges from the states of Amazonas, Acre, Tocantins and São Paulo and from the Federal District were included. All have master’s or doctoral degrees and professional experience as university professors and as technical consultants in the area of infectious and tropical diseases and the epidemiology and clinical treatment of SBEs.

### 3.2. Content Validation

Considering the total number of judges (*n* = 20) and the total scores of the instrument (*n* = 20), a total of 400 possible evaluations were obtained. The content verification index (CVI) score was 90%, 89% and 92% in blocks 1 (objectives of the CPG), 2 (structure and presentation) and 3 (relevance), respectively, thus classifying the CPG as valid in relation to content (Table 2). The item “Is the information well structured in consistency and spelling?”, in block 2, was the only item with a CVI score of <80%. Of these, 215 scores (54%) were totally adequate, 146 scores (36%) were adequate, 33 (8%) were partially adequate and 6 scores (2%) were inadequate.

Major updates were made in version 1 of the snakebite CPG after considering the judges’ recommendations.

### 3.3. Characterization of Target Users

Out of 50 physicians and nurses who attended a meeting for the presentation of the second version of the CPG, 25 (20 nurses and 5 doctors) agreed to participate in the focus groups. Data collection was performed in three focus groups (FG1, 7 participants; FG2, 9 participants; FG3, 9 participants). A purposeful sample of professionals working in health units located in the three municipalities (Careiro da Várzea, 17 professionals; Ipixuna, 4 professionals; and Boa Vista do Ramos, 4 professionals) in the state of Amazonas, western Brazilian Amazon, was included. The participants were aged between 24 and 64 years and had an average of 3 years of professional experience.

### 3.4. Semantic Evaluation by Target Users

During the qualitative analysis, two themes related to semantic validation emerged: *Theme 1*, knowledge acquired in the current meeting, and *Theme 2*, recommendations for the improvement of the care information package.

#### 3.4.1. Theme 1—Knowledge Acquired in the Meeting

In this theme, the quotes indicated new information that the participants had acquired during the CPG presentation. Participants cited incorrect practices in first aid, which they will no longer commit, the advantage of having knowledge about identification of snakes, knowledge of premedication regimens to prevent early adverse reactions, the unnecessary prophylactic use of antibiotics, the importance of knowing the epidemiology for diagnosing the type of envenomation, procedures for wound care, patient follow-up and case reporting. These points are summarized in Table 3.

#### 3.4.2. Theme 2—Recommendations for Improvement of the Care Practice Guideline

Participants recommended inclusions in the CPG that are focused on the antivenom, laboratory tests, concomitant medications and layout of the CPG document (Table 4).

### 3.5. Final Version of the Care Practice Guideline

At the end of the semantics validation, recommendations from the target audience were incorporated into the second version of the CPG. The final version has 46 pages, including a cover page, a summary, a presentation page, 13 chapters (AO1—first aid; AO2—diagnosis and clinical classification of snake envenomations; AO3—Lee–White clotting test procedure; AO4—preparation of antivenom for administration; AO5—antivenom treatment; AO6—wound care; AO7—patient follow-up; AO8—referral of the patient to the higher-level health units; AO9—SBE complications; AO10—hospital discharge criteria and outpatient follow-up; AO11—discharge from the basic unit and follow-up; AO12—case reporting; AO13—receiving and storing antivenoms) and literature references. The document also includes 8 figures, 12 flow charts, 11 tables and the form used for snakebite reporting (Appendix A).

## 4. Discussion

### 4.1. WHO Strategy for Prevention and Control of SBEs and Clinical Utility of SBE Care Information Packages

A CPG for SBE care using community health centers in the Amazon Forest was deemed clinically relevant and valid, as demonstrated by experts’ ratings. This is aligned with the WHO’s strategy for SBE prevention and control which aims to reduce the numbers of deaths and cases of disability by 50% before 2030, improving overall care for patients [4]. In the health system domain, this strategy involves ensuring the production and distribution of safe and effective antivenom treatment and the strengthening of local health systems. The development of a CPG to leverage current healthcare capacity in low-resource settings answers this call.

In Brazil, despite the advances made to achieve self-sufficiency in antivenoms of a high standard and the expansion of a free-of-charge antivenom treatment network, access to SBE treatment is still not available to the most vulnerable parts of the population, which are those that inhabit remote areas of the country, especially in the Brazilian Amazon [3,5]. Strong engagement with communities and health workers is necessary in order to integrate more effective prevention, treatment and management of SBEs into health systems and to encourage victims to seek care early, something which will require a coordinated health promotion strategy [4]. These actions require the development of CPGs for SBE management, such as the one developed in this study. These CPGs must incorporate basic skills and knowledge for pre-hospital care and life support until medical assistance is available and be accompanied by an emphasis on self-management education, as illustrated by our thematic analysis.

From a perspective of decentralizing antivenom treatment to community health centers located in remote areas, the clinical utility of these culturally validated care information packages is even more evident, and it is essential to standardize the care provided to patients. Even in community health centers that do not perform antivenom treatment, it is important to have professionals trained in first aid and the preparation of the patient for transport. However, in general, official guidelines focus almost exclusively on antivenom choice and dosage, without mentioning supportive care. In this study, the panel of expert judges and the target audience approved the fact that the care information package has included this content, which is not usually included in training. In the context of many community health centers, where antivenoms are currently not available, we adapted the CPG, namely in the sections that focused on first aid, wound care and referral of the patient to the higher-level health units, in order to provide users with treatment solutions that are practical for this level of assistance, painkillers and primary wound care, which could provide more time for emergency cases and subsidize comfortable transportation to higher-level health units.

### 4.2. SBE Care Practice Guideline to Improve Care Practice, Add New Knowledge and Reduce Insecurities

In general, physicians and nurses assessed themselves as having very poor knowledge of SBEs, thus resulting in insecure actions and even utilizing incorrect patient care practices. In other situations, professionals were concerned only with referring patients to the nearest hospital units as quickly as possible, in an attempt to free themselves from responsibility. In these cases, most of the time, no care was taken except to activate the available means of transport. It is possible to infer from the professionals’ statements that they recognize that the antivenom is essential for the SBE treatment, but no other first-aid procedure is understood to be necessary to alleviate the patient’s suffering.

Healthcare professionals have doubts as to the effectiveness of non-evidence-based traditional practices, such as those used by the indigenous peoples and riverine populations in the Amazon. Therefore, in the CPG, it was necessary to inform health professionals that some of these practices are ineffective or even harmful (such as tourniquets and incisions), or have unproven effectiveness (such as plant-derived preparations), though at the same time taking care to respect the culture and traditional knowledge of the populations. Some of this information was clearly unexpected by the professionals, who highlighted the need to add these clarifications to the CPG. Participants attributed the low level of knowledge and insecurity to the lack of training they received during their undergraduate courses or training before being referred to work in areas of occurrence of SBEs. In the Brazilian Amazon, the lack of proper training for health workers also results in patients’ resistance to care. Even when antivenom is available locally, some patients may perceive that the quality of care will be better at more distant units and travel very long distances from home (to Manaus, for example) [14].

Health professionals in remote settings may feel insecure about treating SBEs, due to the fear of not being able to manage antivenom-associated early adverse reactions [3]. Knowledge about SBE management, antivenom use and management of antivenom-associated adverse reactions is often poor [7,8]. Developing new or improved treatment guidelines and supporting training programs for health workers are key steps for improving outcomes. In addition, the burden of early adverse reactions must be demystified, principally in relation to frequency and severity [15,16]. In Brazil, about 11% of patients present some type of early adverse reaction, albeit of mild intensity, though these generally allow the antivenom to be given at full dosage [16]. Having information in a document that has detailed procedures and illustrations and workflows to rely upon when an SBE patient is received at the health facility was considered an important way to mitigate insecurities and reduce risks to the patient. In this aspect, nurses clearly understood that they have a crucial role in care, but that no CPG has been developed with a focus on nursing care. Some reported in the focus groups that, in possession of this CPG, they could independently manage the cases of SBEs, since they are continuously at the community health centers, unlike the doctors. Indeed, in Ecuador and Tanzania, successful management of SBEs was achieved in resource-constrained areas by improving access to treatment in nurse-led clinics [17,18].

As for the topic of disseminating knowledge, the participants demonstrated a great interest in propagating all the items of the CPG to the other professionals at their institutions, as well as to the community through health education. This needs to be instituted as a polity action within each municipality so that it is disseminated, known and put into practice by all the actors that make up the health secretariats and their basic health network centers.

### 4.3. SBE Care Information Package Improving Rational AV Use and Reducing Preventable Losses

In Brazil, irrational use of antivenoms is common, with a high frequency of both over- and under-prescribing. Snakebites that do not require antivenom treatment, such as dry bites, in which no venom is injected [19], and bites caused by snakes of no medical importance [20], are common. For the professionals who participated in the study, however, these facts seemed new, as they did not know that there could be cases of snakebites that need only to be placed under observation, without any intervention except for wound care and the use of painkillers in some cases. In these cases, the mistaken administration of antivenom has no clinical benefit to the patient, but may still potentially lead to adverse reactions [19]. Conversely, in the Brazilian Amazon, 52% of patients with severe *Bothrops* envenomations (~90% of the snakebites in the region) were underdosed [19]. In this region, increased lethality was significantly associated with a lack of antivenom administration and antivenom underuse [21,22].

In Brazil, an excessive rate of preventable losses still occurs as a result of there being no specific training in the storage and administration of antivenoms and their expiration due to a lack of proper inventory control [5]. In the first version of the care information package, no specific section was planned in order to deal with the procedures for diluting and administering the antivenom, as it was believed that this knowledge was already in the domain of the nursing team. However, several professionals stated that it was necessary to add this to the final version of the CPG, including illustrations, and this was subsequently done. A specific chapter was also added to deal with the reporting of SBEs since the replacement of the stock of AVs by the state departments is only done based on epidemiological surveillance.

Another common practice reported by professionals was the preemptive use of antibiotics. This intervention is not supported by clinical trials [23] and ends up leading to unnecessary costs for the local health units.

### 4.4. Strengths and Limitations

This study has methodological strengths and limitations that should be considered when interpreting the results. Firstly, evaluations by the judges and health workers about the usefulness, benefits and drawbacks (i.e., the clinical utility) of the CPG are probably influenced by the academic judges and the practitioners’ current work routine and their working context. The expert judges have experience in the field, high-level academic knowledge and vast professional experience, which indicates the extent of the degree of expertise appropriate to the validation of the protocol. Another significant point in relation to the profile of the expert judges concerns their origins, since they work in the northern, southern-central and southeastern regions, which provided diverse and contextualized contributions, thus enriching the inferences given by the professionals in relation to the subject of the study. However, most professionals who participated in the focus groups had little experience in managing SBE patients. If on one hand this prevents them from giving suggestions with a greater technical background, on the other hand these professionals demanded detailing of all the procedures, with illustrations, workflows and improvement in writing of the texts.

Secondly, no gold standard has yet been introduced for instruments used to study the concept of ancillary treatments for SBEs. In general, all the guidelines only focus on AV. Thus, comparison with other instruments relating to other measurement properties, such as cross-cultural and criterion validity, is limited. Thirdly, the recruitment process of study participants was challenging due to the work pressure of the expert judges and the health workers.

## 5. Conclusions

In order for people to be successfully treated and recover from SBEs, they require access to good-quality antivenoms and other care procedures that may be necessary [16]. In Brazil, however, health workers receive little if any training in diagnosing and treating SBEs and many have limited opportunities for in-service professional development of clinical practice. This validated CPG should be integrated into health workers’ curricula at all levels in order to improve SBE management and to increase their knowledge and practical clinical competence. Finally, our findings support the use of this validated CPG in multiple study environments and cultural contexts in Brazil.

## Figures and Tables

**Figure 1 toxins-14-00376-f001:**
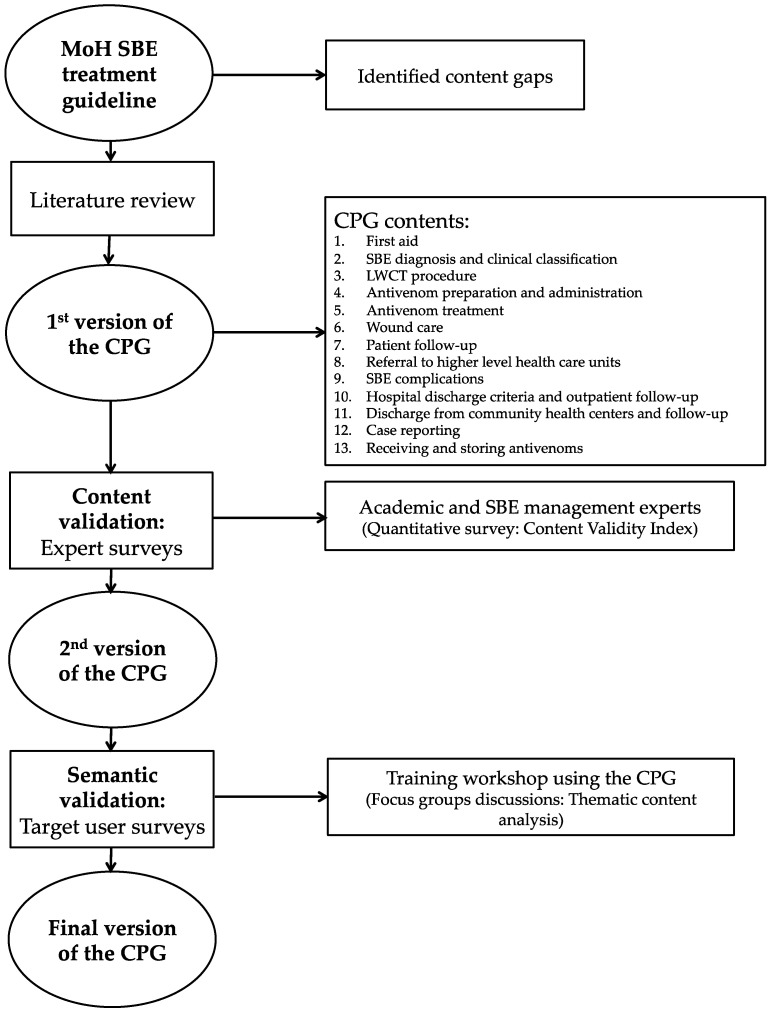
Methods overview for development and validation of the clinical practice guideline (CPG). MoH: Brazilian Ministry of Health; LWCT: Lee–White clotting time test.

**Table 1 toxins-14-00376-t001:** Eligibility criteria for the care information package validation committee.

Criteria	Description	Points
Group 1—judges with academic expertise		
Academic titles	Doctorate in the subject area	3
	Master’s degree in the subject area	2
	Specialist/medical residence in the subject area	1
Professional experience	Minimum of 2 years patient care in the subject area	2
	Minimum of 2 years teaching in the subject area	2
Scientific production	Dissertation, thesis or monography in the subject area	1
	Papers published related to the specific area	2
	Supervision of scientific works in the subject area	1
Group 2—judges with patient management expertise		
Academic titles	Doctorate or Master’s degree in the subject areas	2
	Specialist/medical residence in subject areas	1
Professional experience	Minimum of 2 years patient care in the subject area	2
Additional training	Participation in training courses in the subject area	1
	Attendance of scientific events in the subject area (congresses, meetings, conferences or others)	1

**Table 2 toxins-14-00376-t002:** Content validity index scores for the validation of the snakebite envenoming care practice guideline.

Items	Fully Adequate	Adequate	Partially Adequate	Inadequate	Content Verification Index (%)
1. Block 1—Objectives					
1.1—The content is consistent with the daily needs of the target audience (health professionals) who will use the care practice guideline (CPG).	12	5	3	0	85
1.2—The content is important for the quality of the work of the target audience that will use the CPG.	12	7	1	0	95
1.3—The content instigates changes in the behavior and attitude of professionals who will use the CPG.	11	8	1	0	95
1.4—The CPG is adequate for circulation in the scientific community of the area.	12	5	2	1	85
1.5—The content meets the objectives of health institutions assisting SBE cases.	10	8	2	0	90
Block 1 content verification index scores					90
2. Block 2—Structure and presentation					
2.1—Is the CPG appropriate for use by the target audience?	10	9	0	1	95
2.2—Are the messages presented clearly and objectively?	8	9	2	1	85
2.3—Is the information presented scientifically correct?	11	6	2	1	85
2.4—Is there a logical sequence to the proposed content?	11	6	2	1	85
2.5—Is the information well structured in consistency and spelling?	5	9	6	0	70
2.6—Does the writing style match the knowledge level of the target audience?	9	9	1	1	90
2.7—Is the cover page, summary and/or presentation of the information consistent?	13	6	1	0	95
2.8—Are the title and topic lengths adequate?	12	8	0	0	100
2.9—Are the illustrations expressive and sufficient?	11	8	1	0	95
2.10—Is the number of pages adequate?	11	8	1	0	95
Block 2 content verification index scores					90
3. Block 3—Relevance					
3.1—Do the themes portray key aspects that should be reinforced?	15	4	1	0	95
3.2—Does the CPG allow generalization and transfer of learning to different contexts?	10	7	3	0	85
3.3—Does the CPG propose to build knowledge?	12	6	2	0	90
3.4—Does the CPG address the issues necessary for the target audience’s work?	10	10	0	0	100
3.5—Is the CPG suitable for use by the target audience?	10	8	2	0	90
Block 3 content verification index scores					92

**Table 3 toxins-14-00376-t003:** Knowledge acquired during the meeting for validation of the snakebite envenoming care practice guideline according to the participants.

Knowledge Acquired	Participants’ Quotes
First aid	*“[There was a] patient with a snakebite to the leg, but he already came with a tourniquet… we carried out the transfer to the hospital, but now we know. What I learned… avoid using a tourniquet…” (P4, nurse, Careiro da Várzea)* *“This conduct of using a tourniquet or not, whether it’s right or not… was a great help” (P7, nurse, Careiro da Várzea)*
Diagnosis of the type of SBE	*“To find out more about the type of snake, to differentiate… and what type it could be and what type of snakes there are in our Amazon…* Bothrops, *which is practically the one with the highest percentage” (P4, nurse, Careiro da Várzea)* *“I think that the difference between* Bothrops *and* Lachesis. *That was very clear to me…” (P5, nurse, Careiro da Várzea)* *“Knowing how to differentiate between snakes, in order to give the correct antivenom” (P6, doctor, Boa Vista do Ramos)*
Prevention and treatment of early adverse reactions	*“Pre-medication, which nowadays no longer uses injectable promethazine. You know? That’s what helped me.” (P4, nurse, Careiro da Várzea)* *“The patient had a reaction… and we stopped the antivenom. When we know that the patient has stabilized, he has to continue with the antivenom again until complete… but the antivenom had already been discarded” (P21, nurse, Ipixuna)* *“And, if there is a reaction, we can stop giving the antivenom and do it again. And, if [the patient] is having an allergic reaction, what can you use, what can you do with the antivenom, if it continues or if it stops, so we learn what we can do with an adrenaline shot, if you have an anaphylactic reaction, urticaria and continue with the antivenom again” (P25, doctor, Bos Vista do Ramos)*
Wound care	*“Antibiotics, as I’m a doctor in a rural area… I would probably give the antibiotics at the beginning, now I know I shouldn’t.” (P2, doctor, Careiro da Várzea)* *“I’ll have a different view that I didn’t have before, evaluate the injury of this one. Does he have edema? Do he look flushed? Is he hot? Is there bleeding? Things that could have gone unnoticed before.” (P24, nurse, Boa Vista do Ramos)*
Patient follow-up	*“See if the patient has evolved, if they need to be reclassified, if they need to receive more antivenom again. This is also something I will use in my professional life” (P21, nurse, Ipixuna)*
Case reporting	*“The knowledge we acquired here was very great, when I saw the snakes and I couldn’t identify them, so I had difficulty filling out the notification form, because it asks for the amount of antivenom… so this here opened up our minds a lot…” (P16, nurse, Careiro da Várzea)*

**Table 4 toxins-14-00376-t004:** Participants’ recommendations for improvement of the care practice guideline.

Subject	Recommendations
Antivenom treatment	Describe step-by-step how to administer the antivenom: Detail the antivenom dilution; Detail the antivenom administration; Present images of the types of snakebite antivenom available in the country.
Laboratory tests	Describe how to interpret laboratory results: Describe the Lee–White clotting test result; List other laboratory tests to be performed; Include reference values for laboratory tests in the CPG document.
Concomitant medications	Describe how to proceed in case of concomitant medications: Add possible interactions with other medications, such as medication for treating diabetes and high blood pressure; Add medications that cannot be used.
Care information package document structure	Add figures and flow charts in the procedures described in the CPG.

## Data Availability

The data presented in this study are available in this article and Appendix A.

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
