# Peer review of "Validation of a Culturally Relevant Snakebite Envenomation Clinical Practice Guideline in Brazil"

_toxins, 2022, doi:10.3390/toxins14060376_

Round 1

Reviewer 1 Report

I am happy to read an article dealing with this health problem in Brazil, whose objective is to develop and validate a clinical practice guideline (CPG) for the management of BSE throughout Brazil. This guide has been validated by a group of experts. It is a simple and informative document on the status of snakebite management in Brazil. It reads well, and the use of English is good. Only a few minor problems: The authors point out that antivenom is not available in many areas of Brazil where it is most needed. This is a very disturbing and unacceptable situation in this day and age. I recommend more emphasis on this aspect, including the evidence that the lack of antivenom is a very high-risk factor for lethality (e.g., Souza et al. (2018). Toxicon, volume 145, April 2018, pages 15-24). Before acceptance, I would like to have some additional information about the availability of antivenom in Brazil, especially in the Amazon area where the management of snakebite envenomings poses a difficult challenge for the public system of Brazil. The authors also point out: Even when antivenom is available in low-resource areas, health workers are not adequately trained to manage BSE. Why is this so? What antivenoms are available in Brazil? Speaking of training and education for the medical and paramedical staff? I think it is an excellent initiative to train health professionals in snakebite management in antivenoms. What is the focus of this training? Are they specific? Can you give us more details about the training and education? The authors stated that an excessive rate of preventable losses still occurs due to a lack of specific training in the storage and administration of antivenoms and their expiry due to a lack of proper inventory control. What has been done to avoid preventable losses?

Author Response

We greatly appreciate the Reviewer's comments and suggestions.
The reviewer raises important points about the low coverage of antivenom
treatment in the Amazon, the need for training of health professionals and losses
of antivenoms along the chain, which we believe is also in part associated with
the lack of training of personnel in logistical management.
In this version of the manuscript, we address these problems in the last
two paragraphs of the Introduction and along several points in the Discussion..

Reviewer 2 Report

The manuscript “Validation of a culturally relevant snakebite envenomation clinical practice guideline in Brazil” is well written and I commend the efforts that the authors have undertaken to establish a clinical practice guideline (CPG) for snakebite management in Brazil. However, I unfortunately feel that this manuscript is not appropriate for this journal and will not interest most readers. This manuscript would be better directed toward a public health or policy journal. It details the development of a very specific CPG, as even the inclusion of this CPG in the supplemental material is not provided in English.

Author Response

Please see the Editor's comment:

"I respectfully disagree with the assessment to reject. It is a very important topic. However, Reviewer 2 and Academic editor 1 make excellent suggestions that should be implemented. Such as the CPG should be spelled out in full in English in the primary manuscript and then used as a case study on how to design and implement CPG elsewhere in the world. Therefore, it is an excellent model to use for discussion of the much more fundamental considerations. Therefore, I suggest sending it back to the authors to make such changes so that all the details are in English in the primary manuscript and that they are then used to showcase it so that it is globally relevant."

In the new version, we submitted the CPG in English.

Reviewer 3 Report

The topic processed is undoubtedly extremely important and topical.
However, instead of presenting proposed treatment guidelines, the publication focuses on the process of developing them.
Although this process also has informational value for the scientific community, the biggest shortcoming of the work is the lack of the publication of the English version of the established CPG.
The presentation of the English version of the treatment policy and protocol, would attract a much wider interest.
The scientific value of the communication of the CPG, in my view, goes beyond the description of the efforts and methods to develop it.
I would like to suggest the publication of the English version of the CPG, even in the form of a special issue.

From my humble point of view, the most valuable part of the manuscipt is not transleted in English. If the CPG will be translated, I can support the publication of the manuscript, since the CPG itself exhibits a great value for toxicology. The process of establishing the CPG somewhat seems to be less interesting from toxicologycal point of view.

Author Response

We are very grateful to the reviewer for his suggestions.
In this version, we have considered all these comments,
especially including the English version of the CPG.

Round 2

Reviewer 3 Report

I strongly belive that the CPG will be useful for the experts on this field. Unfortunately, I cannot find the English version of CPG in the supplementary material.